# Northern shrimp *Pandalus borealis* population collapse linked to climate-driven shifts in predator distribution

**R. Anne Richards**[1]*, **Margaret Hunter**[2]

**1** Population Dynamics Branch, Northeast Fisheries Science Center, National Oceanic and Atmospheric Administration, United States Department of Commerce, Woods Hole, Massachusetts, United States of America, **2** Division of Biological Monitoring and Assessment, Bureau of Marine Science, Department of Marine Resources, State of Maine, West Boothbay Harbor, Maine, United States of America

* arichardma7@gmail.com

## Abstract

The northern shrimp (*Pandalus borealis* Krøyer) population in the Gulf of Maine collapsed during an extreme heatwave that occurred across the Northwest Atlantic Ocean in 2012. Northern shrimp is a boreal species, and reaches its southern limit in the Gulf of Maine. Here we investigate proximate causes for the population collapse using data from fishery-independent surveys, environmental monitoring, and the commercial fishery. We first examined spatial data to confirm that the decline in population estimates was not due to a major displacement of the population, and then tested hypotheses related to fishing mortality and shifts in predation pressure. Fishing mortality may have contributed but could not explain the magnitude of the decline or the disappearance of pre-exploitable size individuals. Stomach contents analysis and biomass trends revealed no new fish predators of shrimp. However, longfin squid (*Doryteuthis pealeii* Lesueur) was unique among all species in showing time-series biomass peaks during spring, summer and fall of 2012, and spatial overlap with northern shrimp was unusually high in 2012. Longfin squid is a voracious and opportunistic predator that consumes crustaceans as well as fish. We hypothesize that the warmer temperatures of 2012 not only led to expansion of longfin squid distribution in Gulf of Maine, but had differential effects on migration phenology that further increased spatial overlap with northern shrimp. The weight of our evidence suggests that longfin squid predation was likely a significant factor in the collapse of northern shrimp in the Gulf of Maine.

## Introduction

Climate change has led to well-documented changes in marine, terrestrial and freshwater ecological communities stemming from a diversity of processes, including productivity changes, shifts in species distributions, and changes in timing of seasonal events (phenology) [1–8]. These in turn have the potential to alter competitive and predator-prey interactions, with consequences for species dominance, biodiversity and population persistence. In addition, changes in the physical environment (temperature, oxygen saturation, acidity) exert direct

**Data Availability Statement:** All relevant data are within the manuscript and its Supporting Information files.

**Funding:** The author(s) received no specific funding for this work.

**Competing interests:** The authors have read the journal's policy and have the following conflicts: membership in a government advisory board (RAR and MH). This does not alter our adherence to PLOS ONE policies on sharing data and materials.

physiological impacts that can compromise growth, reproduction and survival, even if well below lethal levels [9, 10].

Climate change has been associated with an increasing frequency of extreme climatic events [11–13], a well-documented example of which was a marine heatwave experienced in the Gulf of Maine (GOM) and more broadly in the Northwest Atlantic Ocean during 2012 [13–17]. This event was the most intense in the GOM in over 30 years, and was superimposed upon a longer-term warming trend that was more rapid than warming in > 99% of the world's oceans [18]. Mean sea surface temperature (SST) in the GOM for 2012 was 2°C above the 1982–2011 average, and was above average in all months of the year [14]. Bottom temperatures were also high, especially in summer and fall [15]. In addition, a phenology shift in the establishment of spring temperature conditions reached a new extreme, occurring more than 2 weeks earlier than in the previous 3 decades [19]. The warm conditions continued into spring of 2013, when the transition to spring also occurred earlier than usual [19]. The 2012 marine heatwave had severe socio-economic repercussions for the highly valued lobster fishery in the GOM through temperature effects on migration and molt timing [13, 14]. In the same year, the population of northern shrimp (*Pandalus borealis* Krøyer) in the GOM suddenly collapsed [20]. This species is the target of a locally important fishery and is another species emblematic of coastal Maine. In this paper, we use foundational knowledge of the biology and ecology of northern shrimp and extensive monitoring data available for the GOM to investigate possible mechanisms involved in the population collapse.

Northern shrimp is a boreal species that reaches the southern limit of its distribution in the GOM, where its temperature sensitivity has been well documented [21–24]. Temperature has been linked to northern shrimp growth rates [25, 26], timing of the larval hatch [27], early life survival [28], and recruitment [24]. In the GOM, the species occurs primarily in the relatively cool western portion of the Gulf [29]. Post-larval stages are benthic and exhibit nocturnal vertical migrations, with the exception of brooding females [23, 29, 30].

Northern shrimp are sequential hermaphrodites. In the GOM, they generally reproduce as males at age 2, transform to females at age 3, and reproduce as females at ages 4 and 5 [23, 31] (age assignments based on length and life history stage [32]). During late fall, the brooding females migrate from offshore habitat (~100–300 m [29]) to near-shore coastal areas (<~100 m [29]) where they hatch their brood during late winter and early spring [23, 27, 29]. Juveniles remain in coastal waters for about a year before migrating offshore to join the mature stock [30, 31]. The fishery has been constrained to winter and early spring since 1999 [33], and targets the brooding females in their nearshore habitat. Although the fishery targets egg-bearing females, which are the largest individuals (generally ≥ 22 mm carapace length (CL)), the fishery avoids exploitation of males (future brood stock) that remain offshore during winter. The fishery was closed in late 2013 and has remained closed since [34].

Here we document the collapse of the GOM northern shrimp population during the 2012 marine heatwave and investigate proximate causes for the collapse. Specifically, we consider potential shifts in distribution of northern shrimp, explore changes in predation including appearance of novel predators, sudden changes in predator biomass, predation intensity, and/or spatial overlap with northern shrimp, and evaluate the possible role of fishing mortality.

## Methods

### Data sources

Our primary data sources were bottom trawl surveys conducted in offshore waters in spring, summer, and fall, and inshore waters in spring (Table 1, Fig 1). Spring and fall offshore surveys were conducted by the Northeast Fisheries Science Center (NEFSC) and summer offshore

**Table 1. Meta-data for fishery-independent surveys used.**

| Survey | Region | Depth (m) | Gear | Tow duration and speed | Average number of stations/year |
|---|---|---|---|---|---|
| NEFSC spring/fall[1, 2] | Offshore | ~56–290 | <2009: 36 Yankee net, 12.5 mm mesh codend liner | <2009: 30 min, 3.8 knots | 42 (85 entire GOM) |
| | | | ≥2009: 4-seam bottom trawl, 25.4 mm codend | ≥2009: 20 min, 3 knots | |
| ME-NH spring[3] | Inshore | ~38–172 | Modified shrimp trawl, 25.4 mm mesh codend | 20 min at 2.5 knots | 67 (109 for spatial indicators) |
| ASMFC & NEFSC summer[4] | Offshore | ~50–250 | Modified shrimp trawl, 25.4 mm mesh codend | 15 min. at 2 knots | 50 |

Depth and station statistics are relevant to shrimp habitat area only, except where noted.

[1][35]

[2] [36]

[3][37]

[4] [38].

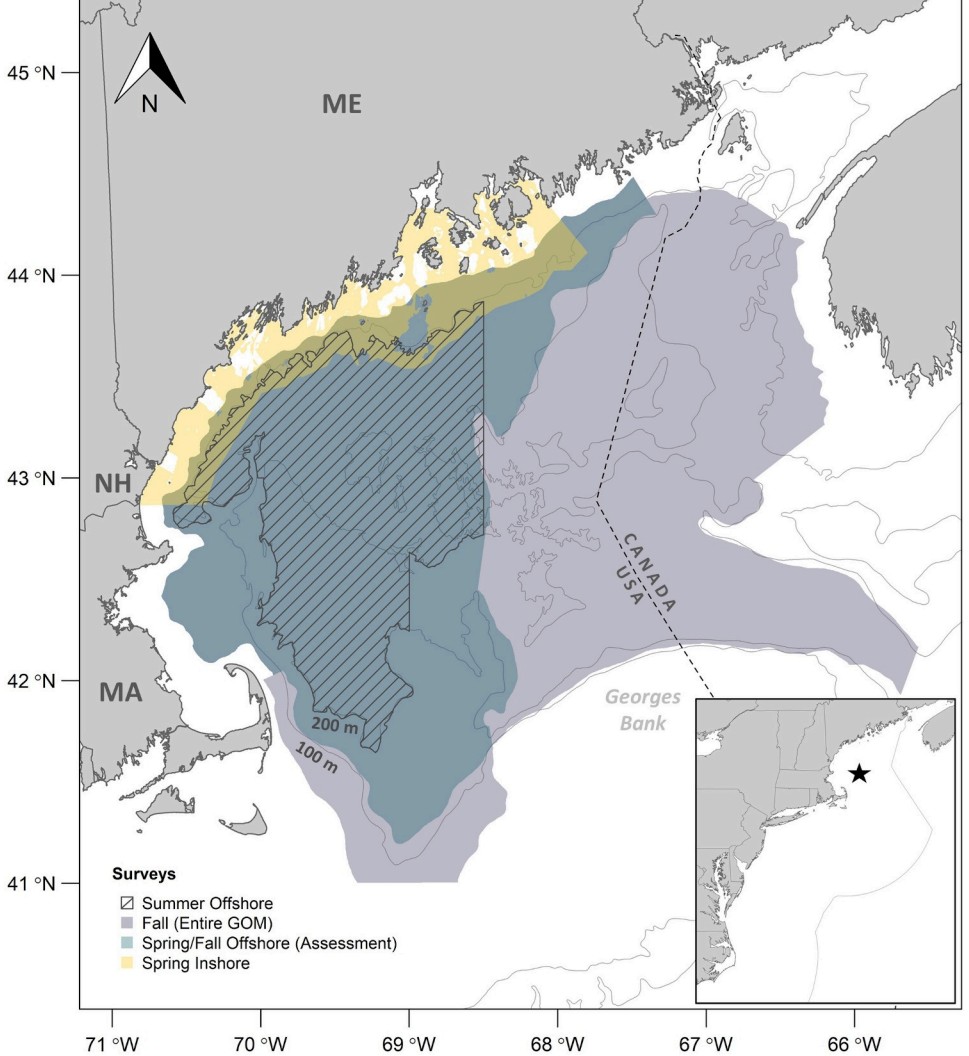

**Fig 1. Map of the study area in the Gulf of Maine, USA.** Inset shows location of Gulf of Maine in the northwest Atlantic Ocean. Credit: Alicia Miller, NEFSC. ME = Maine, NH = New Hampshire, MA = Massachusetts.

surveys jointly by the Atlantic States Marine Fisheries Commission (ASMFC) and NEFSC. The summer offshore survey is a dedicated northern shrimp survey and provides the most complete and detailed data; the other surveys are multi-species surveys which capture a broad array of species. Spring inshore surveys were carried out by the states of Maine (ME) and New Hampshire (NH) and occurred about 3 weeks later than the offshore spring surveys (Table 1). All surveys used stratified random sampling designs (strata defined by depth, latitude and longitude). Biomass and abundance indices were estimated as stratified means, expressed in relative terms because catchability coefficients for many species are either unknown or poorly known. The first year for most analyses was 2003, when the complete set of time series began, and the terminal year was 2017, representing one northern shrimp life cycle (5 years) after the 2012 collapse. The offshore survey regions were the same as those used in annual northern shrimp population assessments (depths 50–290 m) [20], except for analysis of potential shifts to areas outside historical northern shrimp habitat, detailed below. The inshore survey data included depths from 38 m to 172 m except for the spatial overlap analysis (9–172 m). Further details of survey operations are given in Table 1. Sampling conducted as part of this study followed US Government Principles for the care and use of vertebrates (NMFS Animal Care and Use Policy). NMFS fisheries research activities are not subject to IACUC reviews at this time.

Methods for sampling northern shrimp catches were as follows. At each station, the mixed-species catch of caridean shrimp was weighed, and a 2 kg subsample (1 kg in inshore surveys) collected to estimate species composition and sex, life history stage and carapace length composition for northern shrimp. When less than 2 kg (1 kg, inshore survey) of shrimp were caught at a station, the entire catch was processed. Samples were frozen and processed later onshore for the spring inshore and spring and fall offshore surveys, and were processed on board during the summer offshore surveys. Each sample comprised approximately 130 (inshore surveys) to 250 shrimp (offshore surveys) on average.

NEFSC offshore survey methods underwent a major change in 2009 with a new survey vessel, a modified trawl net design, shorter tow duration, and slower trawling speed (Table 1) [39]. Conversion coefficients accounting for the changes were experimentally estimated for many species, and we applied these when calculating survey indices for predators. Conversion coefficients were not available for northern shrimp or some infrequently caught species [40]. For species with no conversion coefficients, NEFSC surveys were treated as two time series, split at 2009. The summer offshore and spring inshore surveys include some fixed (non-random) stations. These were used in analysis of spatial patterns but not in estimating survey biomass or abundance indices.

Shrimp were not identified to species on spring offshore surveys during 2006–2012, therefore we estimated northern shrimp catch for those years from aggregate shrimp catch (all caridean species) using

$$y_i = 0.0133x_i^2 + 0.4849x_i + 0.0249 \quad r^2 = 0.83 \tag{1}$$

Where $y$ = northern shrimp catch, $i$ = tow, and $x$ = total caridean shrimp catch. The relationship was estimated from spring offshore survey data in years when all shrimp were identified (1991–2005, 2013–2016), during which time northern shrimp was 62.3% on average of the total caridean shrimp catch by weight.

All temperature data were collected using conductivity/temperature/depth casts at each survey station. Trends in offshore spring and fall temperatures (bottom temperature, BT and sea surface temperature, SST) were characterized using anomalies that corrected for variation in timing of surveys. The anomalies were calculated relative to a set of reference annual BT and SST cycles that were estimated using measurements taken at approximately 20 fixed sampling

locations in the western GOM 3–6 times per year during 1978–1987 [41]. The anomaly at each survey station was the difference between the observed value and the reference value for that location and date, and the overall anomaly estimate for each survey was the stratified mean of the station anomalies [41]. Comparable anomaly data were not available for the spring inshore and summer offshore surveys, therefore temperature for those series was expressed as stratified means.

Timing of the spring thermal transition was estimated using NOAA's Optimum Interpolation ¼ Degree Daily Sea Surface Temperature (OISST, [19]), calculated for the western GOM (42-44° N latitude, 70.5–68° W longitude, K. Friedland, NEFSC, personal communication). The transition to spring was defined as the first day on which the daily temperature estimate exceeded the average annual temperature [19] during 1982–2017. Duration of summer was taken as the difference between timing of spring and fall onsets (fall onset estimated as the first day on which daily temperature fell below average annual temperature [19]).

## Analysis

**Shifts in distribution.**  To test the hypothesis that the decline of northern shrimp survey indices was due to emigration of northern shrimp to previously unoccupied or less-occupied habitats in the GOM, we examined catch data from NEFSC offshore fall surveys. The fall survey was used because it covers most of the offshore waters of the GOM (Fig 1), and all life history stages (>~age 1) of northern shrimp are distributed offshore in the fall. We calculated two metrics of spatial distribution, the center of gravity (CG) and its inertia or variance, and used these to test for changes in distribution over time.

To estimate CG, distances between points (trawl stations) were calculated in a Euclidean reference system [42]. This was done by setting the minimum longitude and latitude of the strata for each survey as (0, 0) and converting all coordinates to km. The cosine of the midpoint latitude for the respective survey was used to convert longitude. This process is also known as geographical referencing [43].

The CG is the bivariate mean location of the population [42, 44, 45], hereafter referred to as the X- and Y- components of the CG [46, 47]:

$$\text{CG} = \frac{\sum_{i=1}^{n} x_i w_i z_i}{\sum_{i=1}^{n} w_i z_i} \tag{2}$$

where $x_i$ ($i = 1,\ldots, n$) is location (geographically referenced longitude or latitude), $w_i$ is the area of influence, and $z_i$ is the catch biomass (kg). In the case of irregular sampling, spatial indicators are weighted with an area of influence [44, 45]. Given the stratified random survey designs (as opposed to a grid), a Dirichlet tessellation, also known as Voronoï polygons [48], was used as a non-subjective method to calculate areas of influence for each survey, with areas along the edge of the study area clipped to the boundary of the respective strata.

The inertia (variance) describes how dispersed the population is around its CG [42, 44, 45]:

$$I = \frac{\sum_{i=1}^{n} (x_i - \text{CG})^2 w_i z_i}{\sum_{i=1}^{n} w_i z_i} \tag{3}$$

*I* was decomposed into two orthogonal axes describing the maximum and minimum components of the inertia. The square root of *I* for a given axis gives the standard deviation of the respective axis.

Time series analysis of northern shrimp spatial distribution used the same methods as in [46] and [47]. The relationship between each spatial indicator and year was modeled with a linear regression (i.e., a generalized linear model with Gaussian distribution and identity link function). Significant fits were tested for serial correlation; however, no corrections with a first order autoregressive model were necessary.

**Changes in predation.** The potential role of predation was evaluated by investigating whether new predators were present in 2012 and whether predation by known predators had changed in intensity due to increased predator biomass, increased consumption of northern shrimp, and/or increased spatial overlap between northern shrimp and predators.

To identify potential new predators and evaluate whether known predators had anomalously high biomass during 2012, we estimated biomass indices for all species caught in the four seasonal surveys. We took this broad approach to looking for potential predators because shifts in species distributions have been documented on the Northeast US continental shelf in recent years [49–52], suggesting that novel predators could have been present. The analysis excluded bivalve mollusks, American lobsters *Homarus americanus* Milne Edwards, and rock crabs *Cancer irroratus* Say in the offshore survey, and bivalve mollusks, crabs, sponges, echinoderms, and barnacles in the inshore survey, as well as species caught in fewer than 10 tows during the 15-year time series; all other species (n = 59–99 species) were included. We standardized the biomass indices to range between 0 and 1

$$B_{is} = \frac{b_{is} - b_{min_s}}{b_{max_s} - b_{min_s}} \tag{4}$$

where *B* is the standardized biomass index, *i* is year, *s* is species, *b* is the observed biomass index, *min* is time series minimum biomass index and *max* is time series maximum biomass index. We selected species for further consideration if they had a standardized biomass index above 0.85 in 2012. Analysis of spring offshore and fall offshore surveys initially included only 2009–2017 because calibration coefficients were not available for all species caught; however, we found that all species with $B_{is} > 0.85$ had calibration coefficients, so we subsequently used the entire time series (2003–2017). We excluded 10 species with catch rates of $\leq 2$ individuals per survey. We also queried stomach contents data collected from finfish during offshore spring and fall surveys (33 species on average) to identify any new fish species consuming Pandalid shrimp in 2012.

To assess potential changes in intensity of predation on northern shrimp, we examined occurrence of Pandalid shrimp in stomachs of 20 northern shrimp predator species [52] using two metrics: percent of shrimp in the diet (by weight, PW) and percent frequency of occurrence of shrimp in predator stomachs (PFO), excluding species with sample sizes less than 40 stomachs (average) per season. We defined outliers in the PW and PFO time series as observations greater than the mean + 2 standard deviations (SD). The food habits data were collected during NEFSC spring and fall offshore surveys during 2003–2015. Stomach contents were identified from a length-stratified subsample of individuals at each survey station up to a maximum number per species, stratum and station. Data collected included prey species identification, volume or weight of each prey species, prey number, and total stomach volume or weight. To identify new predators, we searched for all species consuming members of the Pandalid family, which could include 4 shrimp species, the biomass of which is dominated by *P*.

*borealis* (Eq 1; [53]). Further detail on food habits sampling and statistical estimators is available in [54]. Diet data were not collected during inshore surveys or summer offshore surveys, and were not collected for invertebrates on any surveys.

Based on results of our predation analyses and existing literature [53, 54], we focused on eight species for investigation of spatial overlap between predators and northern shrimp (spiny dogfish *Squalus acanthias* Linnaeus, Acadian redfish *Sebastes fasciatus* Storer, silver hake *Merluccius bilinearis* Mitchill, Atlantic cod *Gadus morh*ua Linnaeus, white hake *Urophycis tenuis* Mitchill, Atlantic mackerel *Scomber scombrus* Linnaeus, windowpane *Scophthalmus aquosus* Mitchill, and longfin squid *Doryteuthis pealeii* Lesueur). Two metrics were used to evaluate changes in spatial overlap. The first was the proportion of tows that caught both northern shrimp and a given predator, reflecting relatively fine-scale (local) distribution patterns. The second metric was a global index of collocation (GIC), which describes the degree of overlap in the distributions of two populations by comparing the distance between their CGs (ΔCG) and the mean distance between individuals taken at random and independently from each population [44, 45]:

$$GIC = 1 - \frac{\Delta CG^2}{\Delta CG^2 + I_1 + I_2} \tag{5}$$

where *I* is from Eq 3. GIC ranges between 0, in the extreme case where each population is concentrated on a single, but different location, and 1, where the two CGs are identical. GIC values > 0.8 are considered highly collocated, while values between 0.6 and 0.8 are considered to have low collocation [42].

The spatial overlap metrics were estimated for 2003–2017 for each survey. For the offshore surveys, the strata sets were the same as those used in the annual northern shrimp assessments (Fig 1). For the inshore survey, additional strata sets that extended further inshore (~9-37m) and offshore (101–172 m) were included. The crenulated Maine coastline caused incomplete boundaries if the shallower stratum was not included.

**Overfishing.** The potential role of fishing pressure was evaluated using results of a recent benchmark assessment that validated 3 population models developed for GOM northern shrimp [33], as well as a simple model-free analysis developed here. We calculated biomass-weighted relative F as:

$$F_{rel_i} = L_i / SSB_{i-1} \tag{6}$$

where *i* = year, *L* = landings (kt), *SSB* = spawning biomass index from the summer offshore survey [33]. We used SSB indices because these would reflect the exploitable stock available to the fishery in the subsequent year (~5–6 months after the survey). Landings of northern shrimp are considered equivalent to catch because discarding rates of northern shrimp are very low [33, 55]. The full assessment time series (1985–2017) was used for the $F_{rel}$ analysis.

## Results

### Northern shrimp population trends

Biomass indices for northern shrimp declined by over 50% (averaged over all surveys) in 2012 (Fig 2, data in S1 Fig), and abundance indices were very low for all sizes and life history stages (age 1+, Fig 3, data in S2 Fig), including small individuals (primarily males) not exploited by the fishery. Most of the decline occurred between the time of the spring offshore survey (midpoint day of year (DOY) 116, 29% drop in biomass index from 2011) and the summer offshore survey (midpoint DOY 217, 71% drop from 2011). In spring of 2013, the offshore biomass

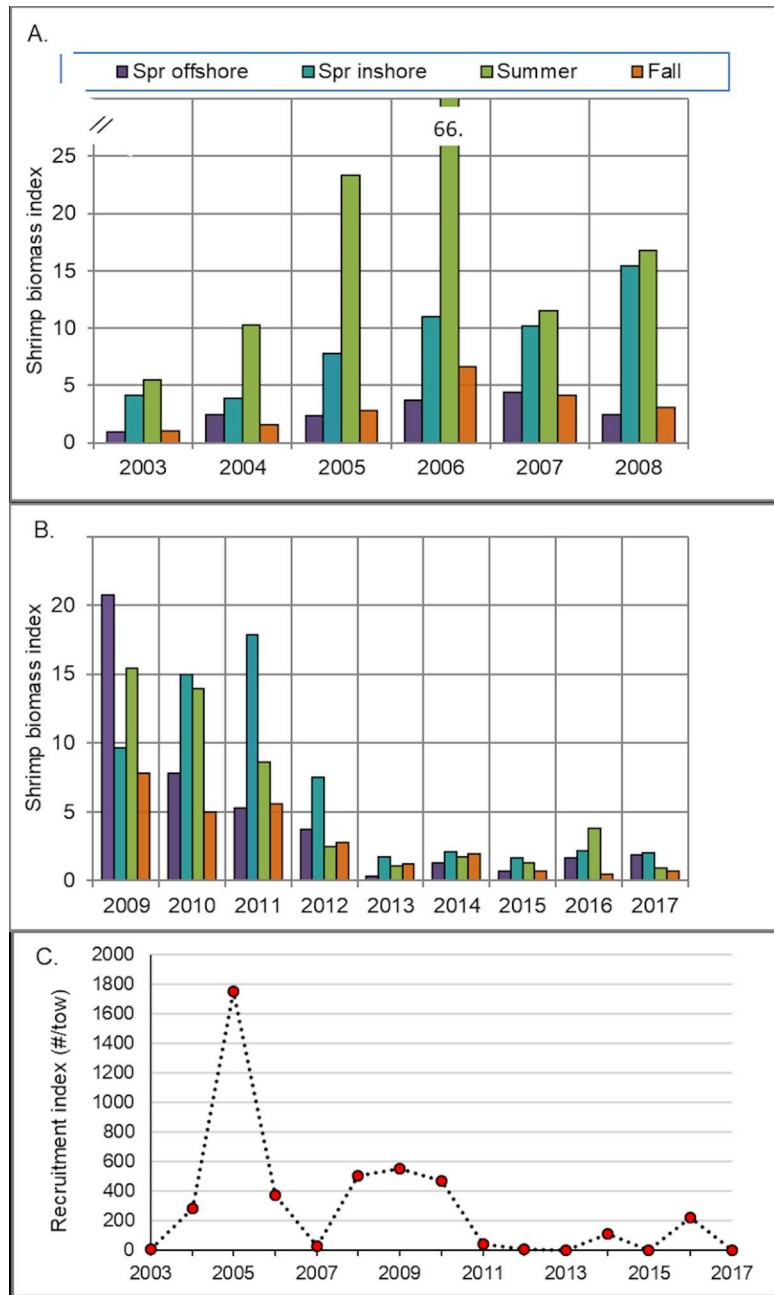

**Fig 2. Relative biomass and recruitment indices for northern shrimp from surveys in shrimp habitat areas.** (A.) 2003–2008, (B.) 2009–2017, (C.) recruitment indices (number per tow of presumed age 1 northern shrimp in summer offshore surveys) [33]. Survey methods for offshore spring and fall surveys changed in 2009. Average coefficients of variation (CV) for the biomass indices were spring inshore 8%, summer offshore 7%, fall offshore 16% (2003–2008), 17% (2009–2017); CV for spring offshore survey was not calculated. Spr = spring.

index was again lower (92% lower than in spring 2012) and all other seasonal surveys were similarly depressed compared to 2012 (spring inshore -77%, summer offshore -58%, fall offshore -56%), suggesting that a further decline had occurred between fall 2012 and spring 2013 surveys.

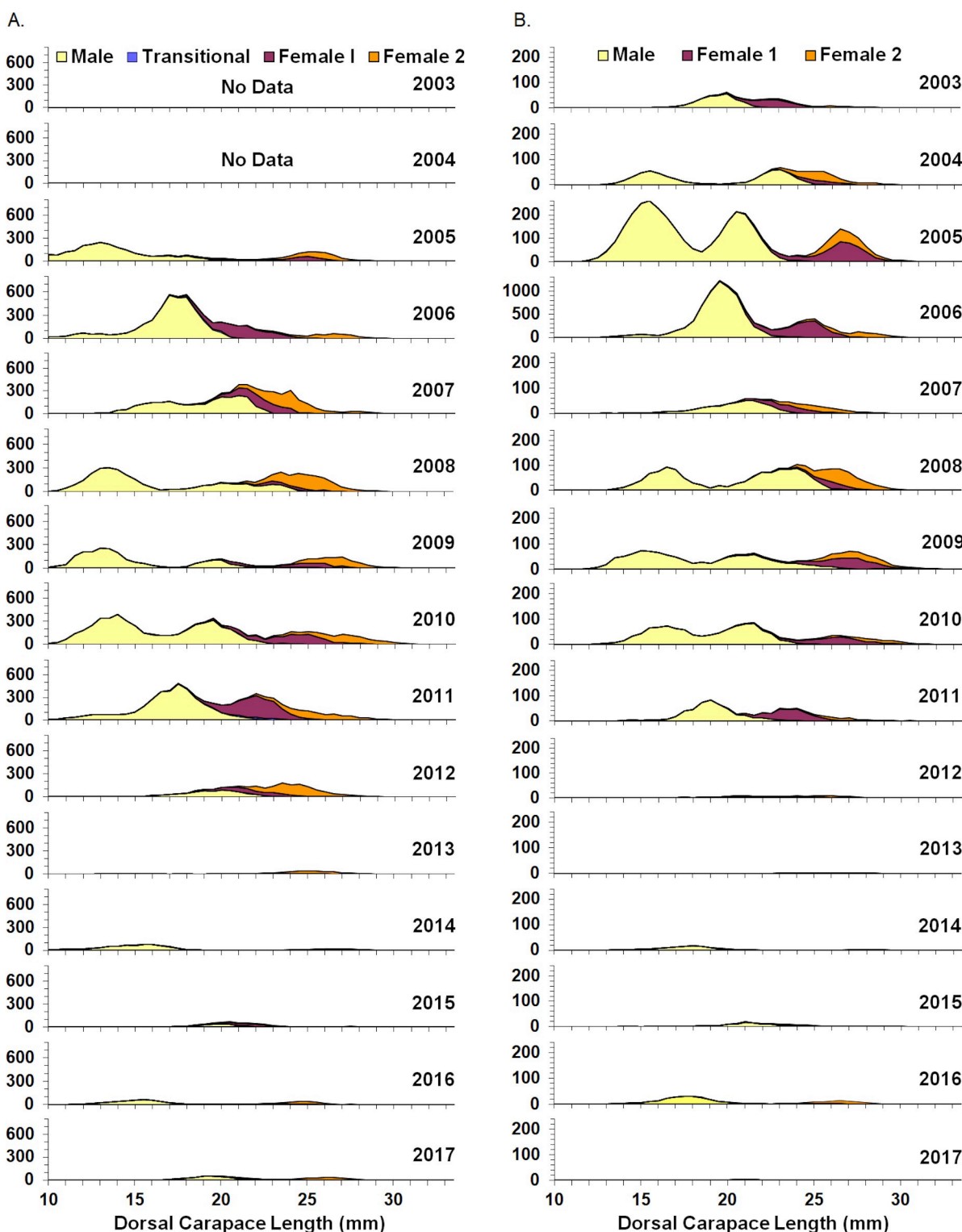

**Fig 3. Stratified mean number per tow at length of northern shrimp by life history stage.** (A.) spring inshore surveys, (B.) summer offshore surveys. Note change in summer offshore y-axis scale in 2006. Female 1 is first year breeding female, female 2 is second year breeding female, transitional is in process of changing from male to female. Life history stage data were not collected during spring inshore surveys until 2005.

## Temperature trends

Offshore spring and fall bottom temperature anomalies (BTA) increased from a low in 2004 to highs in 2011 (spring) and 2012 (spring and fall, Fig 4A., data in S3 Fig). The sea surface temperature anomaly (SSTA) was also relatively high in spring of 2013. Thereafter, temperature anomalies fluctuated at a relatively high level without apparent trend. Spring inshore and summer offshore BT followed a similar trend, with lows in 2004 and highs in 2012 (Fig 4B., data in S3 Fig). Other years with relatively warm BTA and BT in all seasons were 2006 and 2016. The timing of surveys is shown in Fig 4D. (data in S3 Fig).

The onset of spring conditions in the northern shrimp habitat area occurred steadily earlier between 2003 and 2013, shifting at a rate of -2.5 days per year (linear regression, $a = 162.7$, $b = -2.48$, $r^2 = 0.86$, Fig 4C.). The earliest spring onsets occurred in 2012 and 2013, when spring arrived more than 3 weeks earlier than in 2003 (May 15, 2012 and May 13, 2013 vs. June 9, 2003). The duration of summer increased by 4.7 days per year during the same time period (linear regression, $a = 147.9$, $b = 4.70$, $r^2 = 0.87$, Fig 4), with the longest summer (in 2012) eight weeks longer than in the cool year of 2004.

## Shifts in distribution

There was no evidence that northern shrimp distribution had shifted to areas of the GOM outside the historical habitat area. Neither latitude (YCG) nor longitude (XCG) of northern shrimp distribution within the entire GOM showed a significant trend (linear regression,

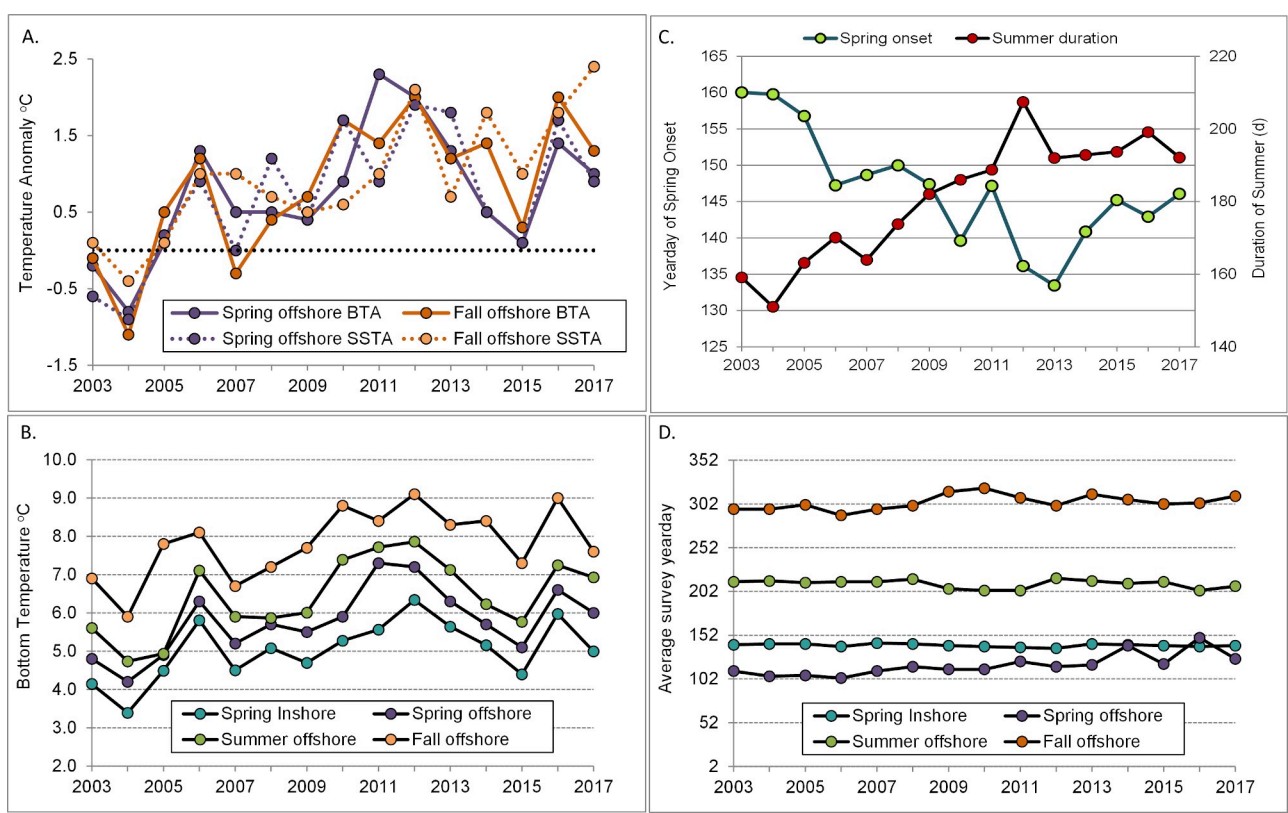

**Fig 4. Bottom and surface water temperature and seasonal phenology for the study area.** (A.) Bottom and surface temperature anomalies (BTA and SSTA, respectively) for NEFSC spring and fall surveys. Anomalies correct for variation in timing of surveys; (B.) stratified mean bottom temperature for spring, summer and fall surveys; (C.) estimated yearday of spring onset and duration of summer in the western Gulf of Maine; (D.) average survey date (yearday).

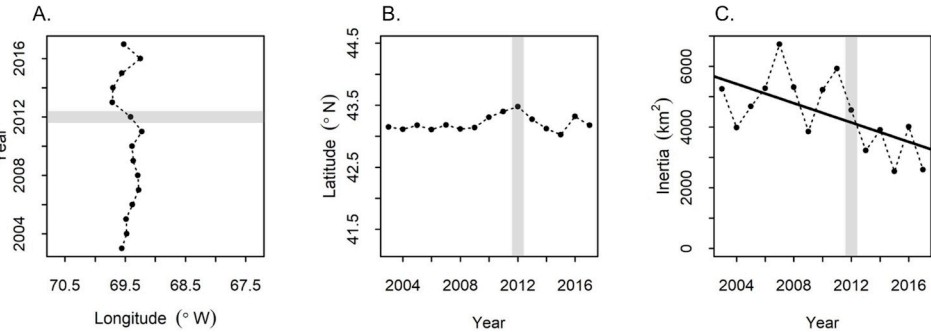

**Fig 5. Results of spatial analysis showing no trend in location but significant contraction of occupied area.** Changes over time in (A.) longitude (back-transformed from XCG), (B.) latitude (back-transformed from YCG) and (C.) inertia (variance of center of gravity) for northern shrimp in NEFSC fall surveys during 2003–2017. Shaded bar highlights 2012, when the shrimp decline became apparent.

P>0.05, Fig 5, data in S4 Fig). The 2012 latitude estimate was the farthest north of the time series but was only 30 km from the time series mean and was well within historically occupied areas (Fig 6, data in S5 Fig). Inertia declined significantly over time ($\beta$ = -159.45, $t_{(13)}$ = -2.70, $p$ = 0.018, Fig 5), indicating an overall contraction of the population.

## Changes in predation

An overview of the results of the predation analyses is given in S1 Appendix.

**Predator biomass.** In our analysis of all species caught in the four surveys (S2 Appendix), 11 species exhibited a time-series biomass peak (standardized biomass index >0.85) in one or

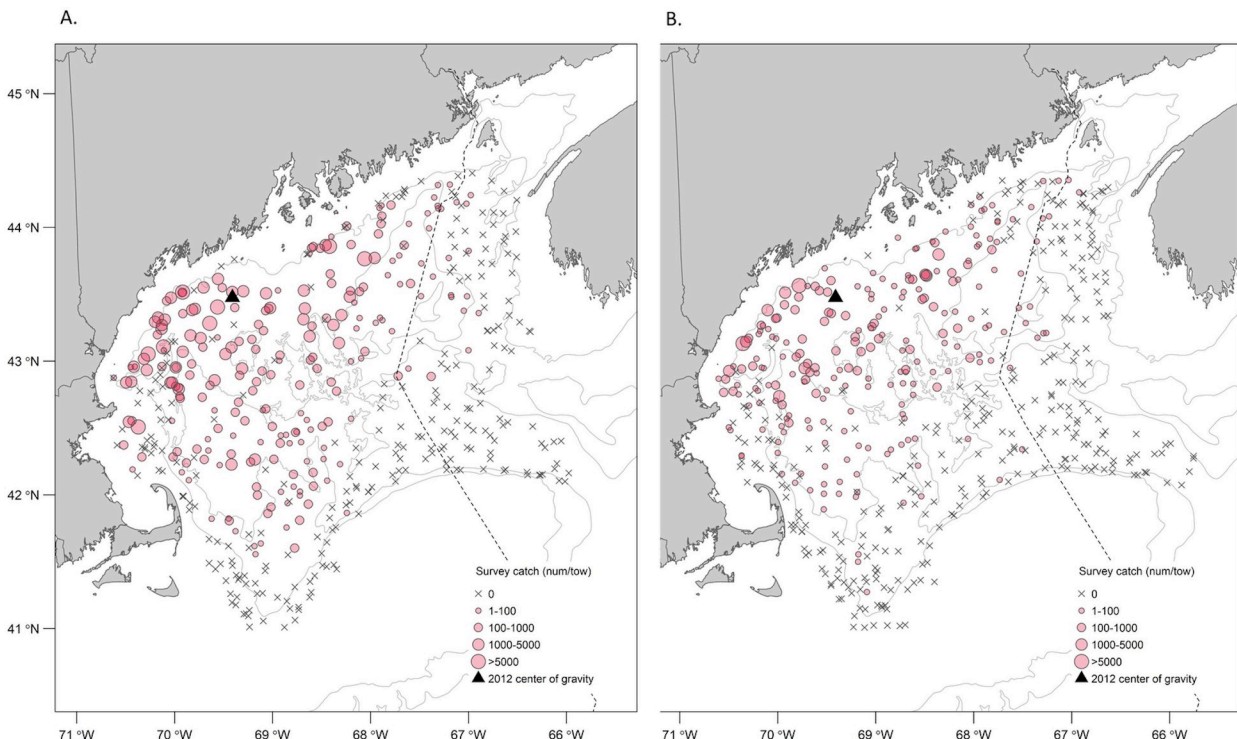

**Fig 6. Number of northern shrimp per tow in fall surveys before and after the population collapse.** (A.) 2006–2011; (B.) 2012–2017. Black triangle marks center of gravity in 2012.

**Table 2. Species with peaks in relative biomass indices in 2012.**

| Species | Spring offshore | Spring inshore | Summer offshore | Fall offshore |
|---|---|---|---|---|
| Atlantic halibut *Hippoglossus* Linnaeus | | x | | |
| Atlantic mackerel *Scomber scombrus* Linnaeus | | | | x |
| Longfin squid *Doryteuthis pealeii* Lesueur | | x | x | x |
| Silver hake *Merluccius bilinearis* Mitchill | | | | x |
| Windowpane *Scophthalmus aquosus* Mitchill | | x | | |
| Atlantic hagfish *Myxine glutinosa* Linnaeus | x | | | |
| Atlantic wolffish *Anarhichas lupus* Linnaeus | x | | | |
| Butterfish *Peprilus triacanthus* Peck | | x | x | |
| American shad *Alosa sapidissima* Wilson | x | | | |
| Gulf Stream flounder *Citharichthys arctifrons* Goode | | | | x |
| Krill (Euphausiacea) | | x | | |
| Number of species examined | 85 | 59 | 75 | 99 |

Gray shaded cells indicate species that are considered unlikely to have had an impact due to diet and/or habitat use.

more seasons during 2012 (Table 2). Five of these species were considered unimportant due to their diet (Atlantic hagfish, Atlantic wolffish, American shad, Gulf Stream flounder, krill) and/or habitat use (butterfish) (Table 2). Of the remaining 5 species, 3 were previously documented predators of northern shrimp in the GOM (Atlantic halibut, silver hake, windowpane; [53, 56]) and the remaining two were Atlantic mackerel and longfin squid. Atlantic mackerel biomass peaked in fall of 2012. Longfin squid was unique in showing biomass peaks in all three seasons in 2012 (in spring inshore, summer offshore and fall offshore surveys), and was also the only species showing a peak in spring of 2013 (offshore). Standardized biomass indices for 20 previously identified predators [53] plus Atlantic mackerel and longfin squid are shown in Fig 7 (data in S6 Fig). The biomass peak offshore in summer 2012 for windowpane was due to a catch of 1 individual, and therefore was dropped from further consideration.

**Predator diets.** No new predators of Pandalid shrimp were identified in 2012 by querying the food habits database, which includes only fish species. Pandalid shrimp did not occur more frequently in stomach contents (PFO) of the 11 fish predators with sufficient sample sizes during NEFSC spring or fall offshore surveys in 2012 or in spring of 2013 (PFO, Fig 8, data in S7 Fig). The only species that was an outlier in terms of percent of stomach content by weight (PW) in 2012 was Atlantic cod in spring (7.4% PW, SD = 3.9, n = 29 tows, 77 stomachs; time series PW mean = 1.6%, average PW SD = 2.3).

Atlantic mackerel was initially excluded from diet analysis due to low sample sizes; however, we examined the stomach content data because of its potential importance indicated by biomass trends. In the 24 stomachs (15 tows) sampled in spring 2012, only one included shrimp-like crustaceans (this broad category could include krill, *Crangon* spp, and other shrimp-like species as well as Pandalids). In fall 2012, neither shrimp-like crustaceans nor Pandalids were found in any of the 26 stomachs sampled (14 tows).

**Spatial overlap.** The proportion of tows catching both northern shrimp and a given predator fluctuated over time, with a drop for some species after the 2012 northern shrimp decline (Fig 9, data in S8 Fig). Silver hake and white hake were caught with northern shrimp in a high proportion of tows, as were Acadian redfish and spiny dogfish in summer offshore surveys. The only species that showed clear increases in tow-by-tow co-occurrence in 2012 was longfin squid (all four seasonal surveys). Co-occurrence with squid remained relatively high in spring of 2013, both inshore and offshore.

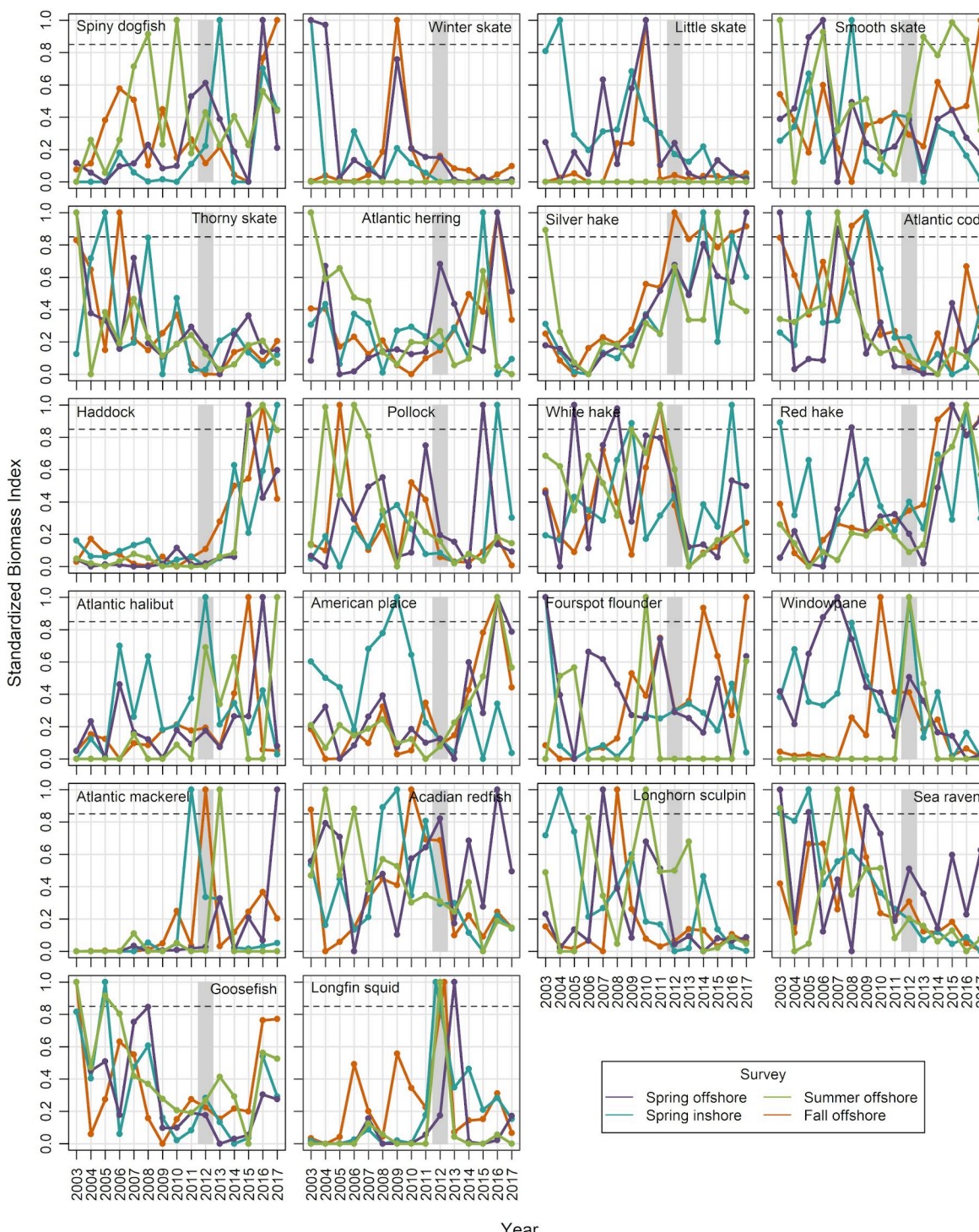

**Fig 7. Standardized relative biomass indices (mean kg/tow) for northern shrimp predators in the western Gulf of Maine.** Indices from offshore spring and fall surveys were calibrated for changes in survey methods beginning in 2009. Shaded bar highlights 2012, when the shrimp decline became apparent. Dashed line indicates standardized biomass = 0.85. Species were selected for further consideration if they had a standardized biomass index above 0.85 in 2012.

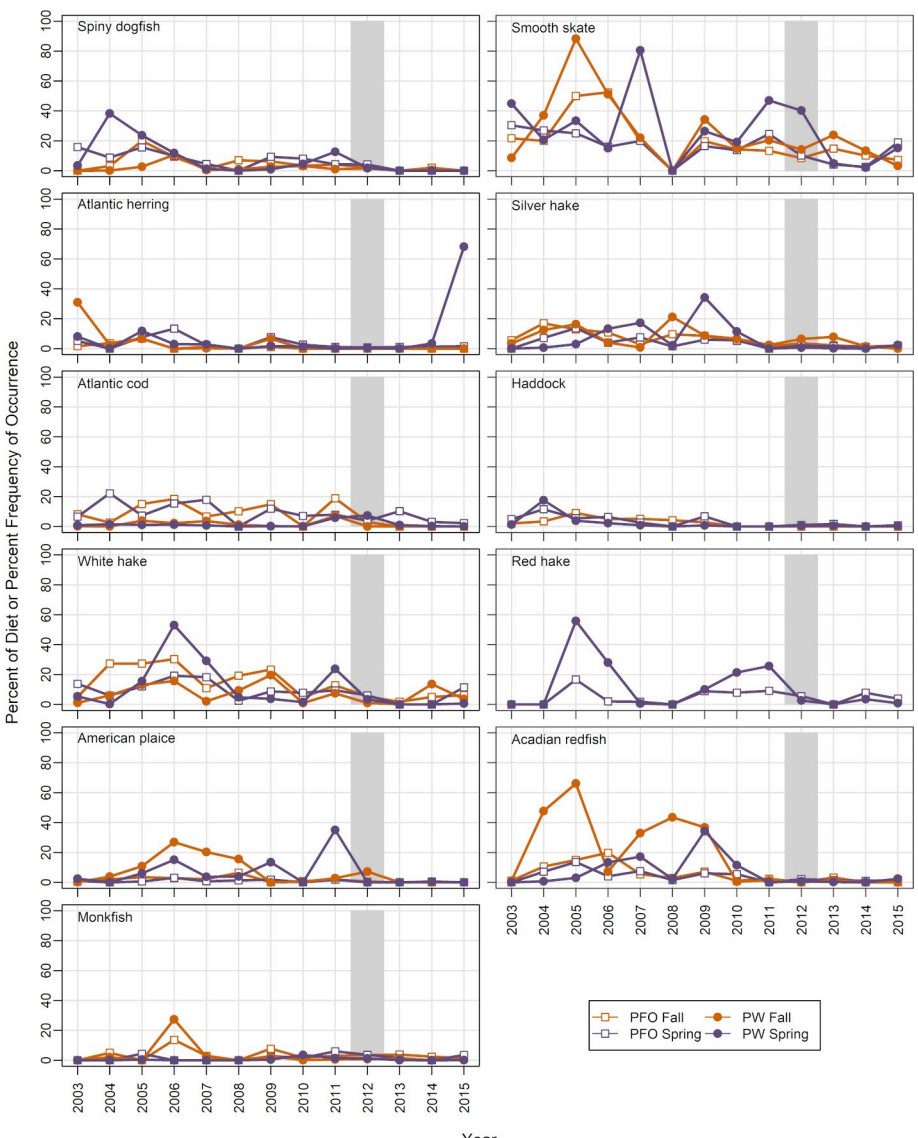

**Fig 8. Percent frequency of occurrence and percent by weight of Pandalid shrimp in fish predator stomachs.** Food habits data were collected in offshore spring and fall surveys. Shaded bar highlights 2012, when the shrimp decline became apparent.

The global index of collocation (GIC) showed similar patterns of overlap (Fig 10, data in S9 Fig), with high collocation of northern shrimp with silver hake and white hake in most seasons and years, and with other species in some seasons (e.g. spiny dogfish, white hake and Atlantic cod in summer). In 2012, GIC was high for Acadian redfish in the spring inshore survey, for spiny dogfish, silver hake, Atlantic cod, white hake and longfin squid in the summer offshore survey, and for Atlantic mackerel and longfin squid in the fall offshore survey. The overlap with longfin squid offshore in summer 2012 was highly unusual. Prior to 2012, the longfin squid collocation index was 0 for the offshore summer survey, but was 0.98 for 2012.

**Overfishing.** Relative F was high in 2011, but not in 2012 (Fig 11, data in S10 Fig). $F_{rel}$ values were highest during 1995–1998, with 1995 $F_{rel}$ more than double the 2011 estimate and four times the 2012 estimate.

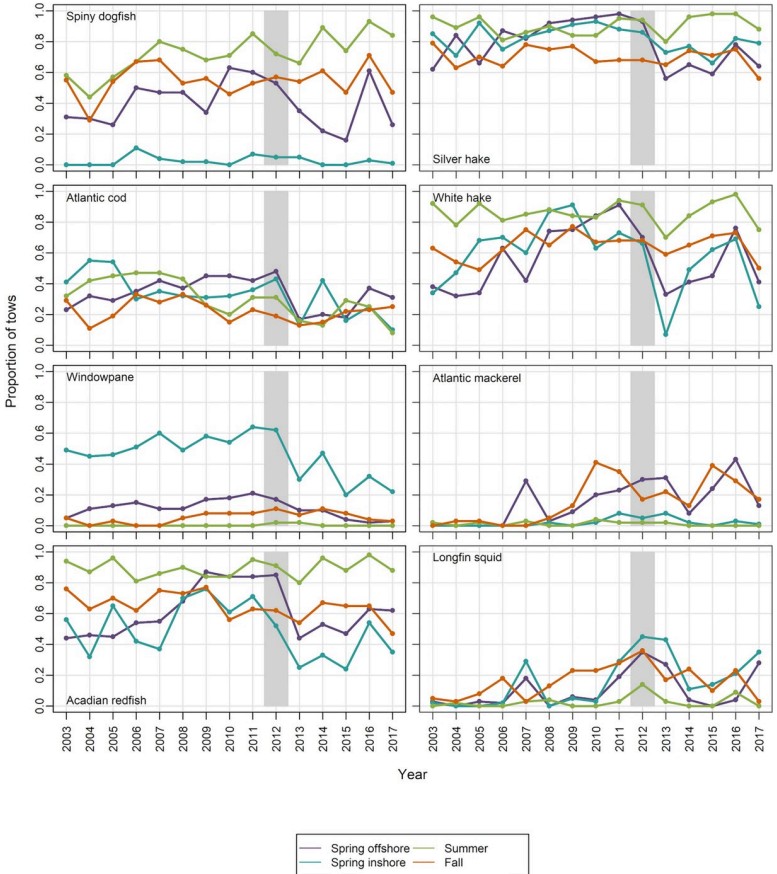

**Fig 9. Local index of collocation (proportion of tows catching both predator and northern shrimp).** Shaded bar highlights 2012, when the shrimp decline became apparent.

## Discussion

During 2012, temperature in northern shrimp habitat areas in the GOM reached record highs, spring conditions were established unusually early, and summer persisted longer than usual. Survey indices for northern shrimp plummeted, affecting both exploitable-size shrimp and shrimp too small to be vulnerable to the fishery. Indices were further depressed the following spring, and the ASMFC declared the northern shrimp population in the GOM collapsed [20]. Despite a fishing moratorium beginning in December 2013, the population had not recovered by 2019 [34].

The 2012 marine heatwave is an obvious culprit in the northern shrimp collapse, given the boreal distribution of this species and the well-documented effects of temperature on its biology [22–25, 27, 28]. However, the potential mechanisms involved are not obvious. Acute physiological effects appear unlikely as a primary cause. Although bottom temperatures reached record highs, they remained 2–3°C cooler than recorded for some breeding populations (11–12°C, [57, 58]), and some evidence suggests that lethal temperature may be as high as 16.5°C [23, 57]). Several northern shrimp populations persist in thermal environments that are warmer than the GOM despite being located further north (e.g. the Skagerrak and Norwegian Deep, Norway and the North Sea [23, 58–60]). More subtle effects of warmer water in the GOM (e.g. declines in metabolic performance [9]) would be difficult to observe, and cannot be fully ruled out. However, in the few studies that have been conducted with juvenile and adult

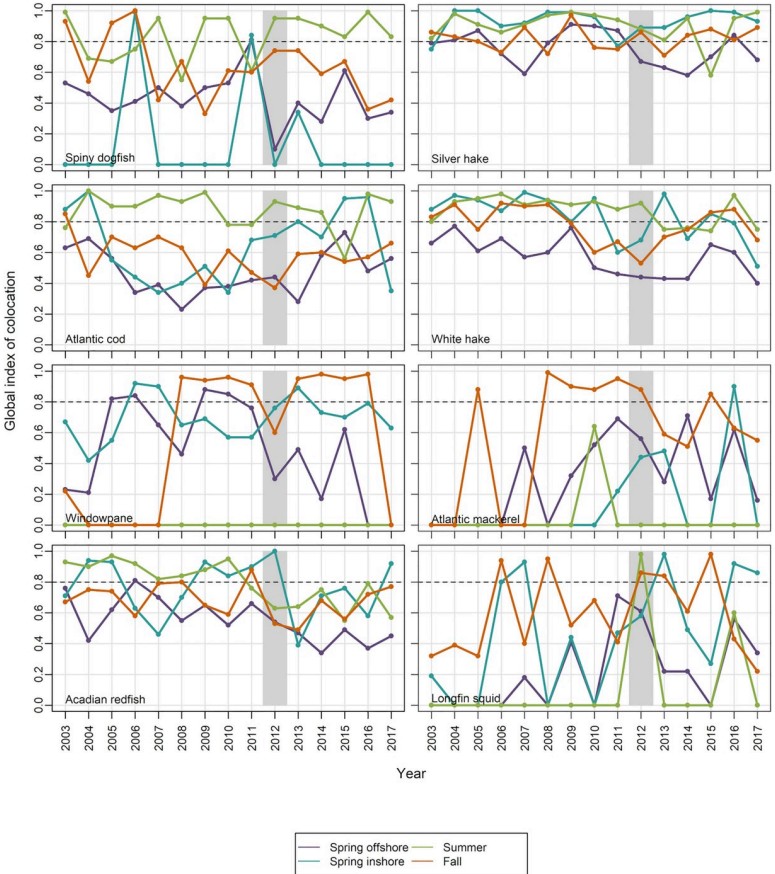

**Fig 10. Global Index of Collocation (GIC) of 8 predators with northern shrimp.** Values above the dashed line (GIC > 0.8) indicate high collocation. Shaded bar highlights 2012, when the shrimp decline became apparent.

northern shrimp, temperature effects on growth (positive, [26]) and physiological condition (negative, [61]) were not observed until treatments were 3–5°C above what would have been encountered in the natural environment where these experiments were conducted [62]. These

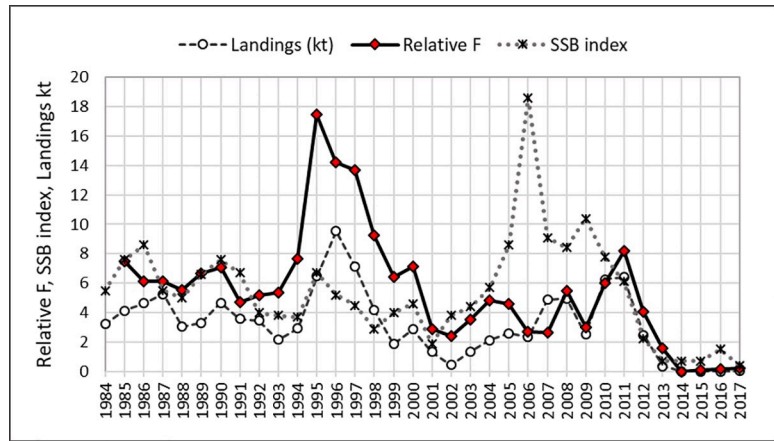

**Fig 11. Northern shrimp biomass-weighted relative F, landings (kt), and spawning stock biomass index.** Spawning stock biomass (SSB) index (based on summer offshore surveys) and landings data are from [34].

results suggest that sub-lethal temperatures were unlikely to have been an immediate cause of the 2012 collapse. Direct physiological effects of high temperatures have rarely been identified as the proximate cause of local extinctions or population declines [1].

The impact of fishing mortality (F) is difficult to quantify, especially in the absence of biological reference points [33]. A recent benchmark assessment estimated very high fishing mortality in 2011 and 2012 [33, 63], suggesting that fishing mortality may have been a factor in the population collapse. However F estimates were also very high during 1995–2001, exceeding the 2011–2012 values in two of the three models evaluated, with no concomitant collapse of the population. Our calculations similarly indicated high relative F in 2011 (though not in 2012), but $F_{rel}$ was much higher in the mid-1990s than in 2011. These lines of evidence suggest that high F could have contributed to, but cannot fully explain, the population collapse seen in 2012. Even if overfishing had been occurring, it could not explain the demise of second-year male northern shrimp because they remain in offshore waters and are not available to the fishery, which is prosecuted inshore.

An alternative explanation for the decline of northern shrimp indices is that the population did not collapse, but instead moved out of its established habitat into waters outside the surveyed areas. Our analysis of spatial distribution within the GOM did not support this hypothesis. The result is unsurprising because bottom waters in the southwestern portion of the GOM are colder than in the northeastern part [29, 52, 64]. While we could not determine whether northern shrimp may have moved entirely out of the GOM (e.g. northeastward to the Scotian Shelf), such movement would require an extensive counter-current migration for which there is no evidence. Genetic studies suggest little to no mixing between GOM and Scotian Shelf northern shrimp stocks [60], despite the potential for downstream drift of Scotian Shelf northern shrimp into eastern GOM. Northern shrimp have not been found in NEFSC annual surveys to the south of GOM (NEFSC unpublished survey data). We conclude that a shift in distribution cannot explain the decline in northern shrimp abundance indices in the GOM, and that the survey indices reveal a true population collapse.

A review of climate-linked population declines and local extinctions found that species interactions were the immediate cause in most studies where mechanisms could be identified [1]. In the case of GOM northern shrimp, changes in several predator-prey relationships could have been involved in 2012 and early 2013. Three previously identified predators had a biomass peak in 2012 in one of the four surveys (Atlantic halibut, silver hake, windowpane). Atlantic halibut has a relatively high average PFO of Pandalid shrimp in its diet (12.5% of stomachs, [53]) and could have been important inshore in spring of 2012. Silver hake generally has a close spatial association with northern shrimp and biomass peaked in fall 2012, but PFO and PW were relatively low (Fig 8). Windowpane has a low average PFO for shrimp (1.4%, [53]) and low spatial overlap, but had relatively high biomass in the inshore spring survey in 2012. However, the species that emerged most frequently in our indicators was longfin squid, which had high biomass in 3 of 4 surveys in 2012 and in spring of 2013, and much higher than usual spatial overlap with northern shrimp (Figs 9 and 10). Longfin squid is a semi-pelagic species that can reach mantle lengths of over 40 cm [65, 66], and occupies seasonally-varying depths ranging from 6 to 400 m. Longfin squid undertake diel vertical migrations, as do northern shrimp, occurring near bottom during the day and higher in the water column at night [67, 68]. Their rapid individual growth and short life cycle (9–12 months longevity [65, 66]) allows the population to respond quickly to changes in conditions. Food habits data for cephalopods are not routinely collected on NEFSC surveys because their food is thoroughly masticated, making at-sea identification of prey species difficult [69]. However, targeted studies have shown that crustaceans, including various species of shrimp, can be an important diet component [67, 70–72]. In a review of longfin squid feeding patterns, Macy [70] concluded

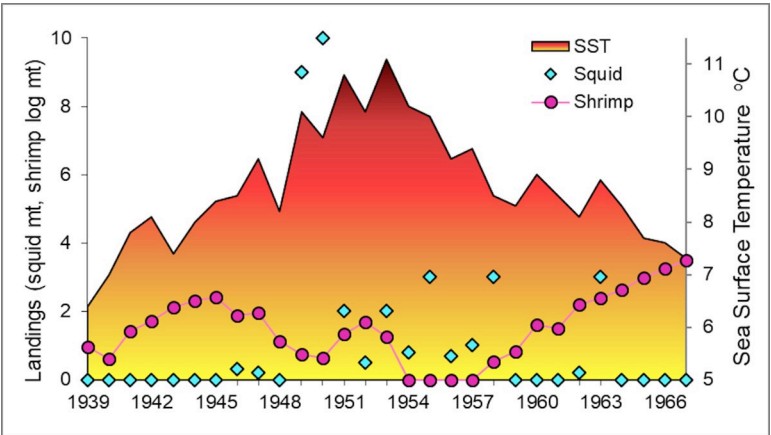

**Fig 12. Maine longfin squid and northern shrimp landings and annual average sea surface temperature.**
Temperature recorded daily at Boothbay Harbor, Maine, 1939–1967. Squid landing units are mt, shrimp landings are in log(mt+1). Data from [21, 77].

that the longfin squid is "a highly opportunistic predator, whose diet primarily reflects the local abundance of potential prey species". These observations leave little doubt that longfin squid has the potential to prey on northern shrimp given the opportunity. Unfortunately we do not have direct observation of longfin squid diets during the collapse period, and laboratory studies to examine feeding behavior of longfin squid were outside the scope of this study.

Could longfin squid have been a major player in the collapse of the GOM northern shrimp population? Squid species are frequently identified as keystone species, i.e. those capable of exerting a strong effect on ecosystems even at relatively low biomass (e.g. [73]). In the Gulf of California, a major drop in sardine landings was associated with an influx of Humboldt squid *Dosidicus gigas* in the previous year [74], and depressed landings of Pacific hake have been associated with spikes in *D. gigas* abundance further north in CA [75]. A bioenergetics study of longfin squid in the Northwest Atlantic concluded that the species was capable of exerting control over recruitment of the five finfish species that were examined [76]. In the GOM, the historical record provides further evidence for a possible controlling role for longfin squid. During the warm 1950s, northern shrimp catches dropped to zero despite continued fishing effort, and a fishery for longfin squid developed ([21, 77], Fig 12). A similar squid fishery response to the 2012 influx of longfin squid was seen as well [13, 14]. More recently, an extreme drop in northern shrimp abundance indices, including pre-recruit males, occurred contemporaneously with an increase in longfin squid biomass indices during fall 2006 through summer 2007 (Figs 3 and 7). These observations suggest possible trophic control by longfin squid, and align with observations of squid impacts in other systems.

We hypothesize that the impact of longfin squid could have been exacerbated in 2012 due to the early onset of spring. Historically longfin squid have not been abundant in GOM inshore waters until summer. By the time the longfin squid arrive, female northern shrimp would have migrated back offshore from coastal waters after hatching their brood, thus overlapping very little with longfin squid (Fig 13). With the earlier arrival of spring in 2012, the influx of longfin squid apparently occurred earlier (confidential monthly landings data, ME DMR and NEFSC), a response that has also been documented in a related squid species [78]. The northern shrimp hatch period began early in 2012, but its duration was unusually long, ending only 4 days earlier than average [27]. Thus female northern shrimp were likely still available inshore when the squid arrived, resulting in higher than usual spatial overlap in

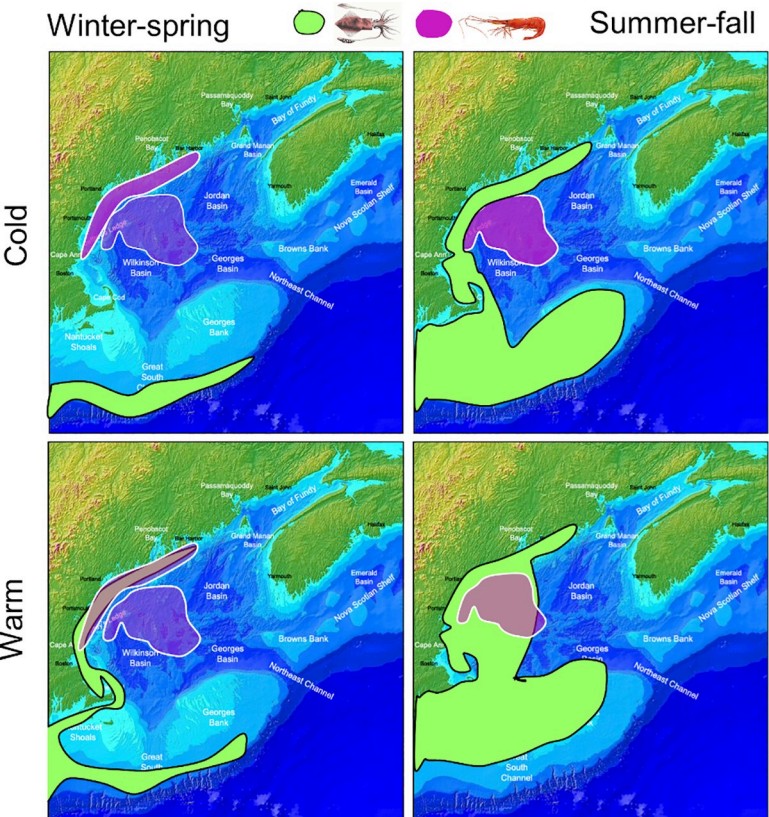

**Fig 13. Conceptual model of seasonal distribution patterns of northern shrimp and longfin squid.** (A.) cold years, (B.) warm years.

coastal areas (Figs 9, 10 and 13). This suggests that mismatched shifts in migration phenology may have increased the opportunity for longfin squid to prey on female northern shrimp in the inshore habitat, in addition to their expanded presence in offshore waters in 2012 and early 2013.

Northern shrimp in the GOM may have been caught in a squeeze between bottom up and top down processes in recent years. The GOM has been warming rapidly [14, 18], and the northern shrimp population was showing signs of stress before 2012 with low survival to age 1 of the 2010 and 2011 year classes [28, 33], which were hatched in relatively warm years (Fig 4). Previous studies of GOM northern shrimp have linked warmer temperatures to a depressed stock-recruitment relationship [24], and to reduced early life survival [28]. These may reflect bottom up pressures such as metabolic tradeoffs or trophic mismatch [28], or top-down pressure on early life stages, possibly in the form of egg parasitism [79] or increased predation. In contrast, the 2012 collapse was sudden as might be expected from an abrupt shift in predation pressure, a top-down effect. A shift in the balance between bottom up and top down pressures may explain why the population quickly recovered from a low spawning biomass in 2001 while it did not in 2012. The years 2001 and 2002 were relatively warm, but were followed by several unusually cold years and the highest recruitment indices on record [33]. The 2012 heat wave was followed by another warm year, a return to more average temperatures during 2014–2015, then another warm year in 2016. Studies of other northern shrimp populations have also shown the importance of both the physical environment and predation in the species' population dynamics [80–84]. Though not as extreme as during 2012 and early 2013, squid biomass

has remained generally higher (Fig 7), temperatures relatively warm in most years (Fig 5) and spring relatively early (Fig 5) compared to the early 2000s.

Factors that we could not address in this study could have played a role in the collapse. For example, climate change is expected to increase susceptibility of crustaceans to disease, particularly when pathogens have optimal temperatures higher than their host's [85]. Two pathenogenic parasites are associated with northern shrimp in GOM, one causing egg mortality [79] and the other causing black spot gill syndrome (BSGS, [86]). Egg mortality would not explain a sudden population collapse, but BSGS causes necrosis of infected gill lamellae, which presumably compromises shrimp condition. BSGS was first recorded for GOM northern shrimp in 1967, with the highest rates of infection (proportion of individuals infected) approximately 55% in fall and winter when females were egg-bearing (i.e. not molting) [87]. In limited sampling during fall 2012, 95% of shrimp showed some level of infection, and the average proportion infected was >70% in 4 of 5 years of limited sampling during 2012–2016 (H.-Y. Chang, Univ. Maine, personal communication). It seems possible BSGS could have contributed to the continued decline of shrimp between fall of 2012 and spring of 2013, but beyond that we are unable to speculate about the role of episodic disease in the collapse.

Other factors frequently cited as detrimental effects of climate change include ocean acidification (OA) and major changes in trophic ecology. Experimental work with northern shrimp larvae reared at pH predicted for the year 2100 showed longer developmental times [88, 89], but survival was not affected [88]. Due to the complexity of the carbonate system in the GOM, recent warming events have not resulted in strong acidification [90], suggesting that OA was not a major factor in the 2012 collapse of northern shrimp. Changes in trophic ecology in the GOM have been observed in recent years (e.g. [6, 91–93]); however, these seem unlikely to have resulted in a catastrophic decline of post-juvenile northern shrimp given the generalist feeding strategy of the shrimp [23].

## Conclusions

Understanding processes occurring in the natural environment is challenging because of the multiplicity of mechanisms interacting and fluctuating through time. Dramatic events such as the GOM shrimp population collapse provide an opportunity to see through the haze a bit more clearly. Our results suggest that longfin squid may have been a major player in the collapse of GOM northern shrimp during an extreme marine heatwave event, and provide further evidence that changing species interactions will have major impacts as ecosystems reorganize due to climate change.

## Supporting information

**S1 Appendix. Summary of results of analysis of changes in predation.** Species scientific names are given in S2 Appendix.
(DOCX)

**S2 Appendix. Common and scientific names for all species captured, 2003–2017.**
(XLSX)

**S1 Fig. Relative biomass and recruitment indices for northern shrimp from seasonal resource surveys in the western Gulf of Maine.**
(CSV)

**S2 Fig. Stratified mean number per tow at length of northern shrimp by life history stage in the western Gulf of Maine.**
(CSV)

**S3 Fig. Survey timing, bottom and surface water temperature, and seasonal phenology in the western Gulf of Maine.**
(CSV)

**S4 Fig. Estimates of mean longitude, latitude and inertia of survey catches of northern shrimp in the entire Gulf of Maine.**
(CSV)

**S5 Fig. Number of northern shrimp per tow in fall surveys covering the entire Gulf of Maine, before and after the population collapse.**
(CSV)

**S6 Fig. Standardized biomass indices (mean kg/tow) for northern shrimp predators in the western Gulf of Maine.**
(CSV)

**S7 Fig. Percent frequency of occurrence and percent by weight of Pandalid shrimp in fish predator stomachs in the western Gulf of Maine.**
(CSV)

**S8 Fig. Local index of collocation (proportion of tows catching both predator and northern shrimp) in the western Gulf of Maine.**
(CSV)

**S9 Fig. Global Index of Collocation (GIC) of 8 predators with northern shrimp in the western Gulf of Maine.**
(CSV)

**S10 Fig. Northern shrimp biomass-weighted relative F, landings (kt), and spawning biomass index in the western Gulf of Maine.**
(CSV)

**S11 Fig. Raw data for calculating statistical indicators shown in Fig 5.**
(CSV)

**S12 Fig. Raw data for distribution map shown in Fig 6.**
(CSV)

## Acknowledgments

Many thanks to Charles Adams for spatial indicator analysis, Alicia Miller and Chris Tholke for technical support, and Kevin Friedland for providing seasonal timing estimates. We gratefully acknowledge insightful comments from members of the fishing industry, and the hard work of survey crews, commercial samplers and database developers that made this study possible.

## Author Contributions

**Conceptualization:** R. Anne Richards, Margaret Hunter.

**Data curation:** R. Anne Richards, Margaret Hunter.

**Formal analysis:** R. Anne Richards, Margaret Hunter.

**Investigation:** R. Anne Richards, Margaret Hunter.

**Methodology:** R. Anne Richards, Margaret Hunter.

**Project administration:** R. Anne Richards.

**Software:** R. Anne Richards.

**Visualization:** R. Anne Richards, Margaret Hunter.

**Writing – original draft:** R. Anne Richards.

**Writing – review & editing:** R. Anne Richards, Margaret Hunter.

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
