## [Decision Letter · Decision Letter 0]

16 Apr 2021

PONE-D-21-06531

Northern shrimp Pandalus borealis population collapse linked to climate-driven shifts in predator distribution

PLOS ONE

Dear Dr. Richards,

Thank you for submitting your manuscript to PLOS ONE. After careful consideration, we feel that it has merit but does not fully meet PLOS ONE’s publication criteria as it currently stands. Therefore, we invite you to submit a revised version of the manuscript that addresses the points raised during the review process.

I tend to agree with the comments/suggestions made by reviewer #1. In particular in your revision try to address in your discussion some reasons/thoughts on why the population has not recovered in recent years; and include also some information on recruitment processes in relation to the stock SSB.

We look forward to receiving your revised manuscript.

Kind regards,

Andrea Belgrano, Ph.D.

Academic Editor

PLOS ONE

Journal Requirements:

"The authors have read the journal’s policy and have the following conflicts: membership in a government advisory board (RAR and MH)"

4. We note that Figures 1, 6, 13 in your submission contain map images which may be copyrighted. All PLOS content is published under the Creative Commons Attribution License (CC BY 4.0), which means that the manuscript, images, and Supporting Information files will be freely available online, and any third party is permitted to access, download, copy, distribute, and use these materials in any way, even commercially, with proper attribution. For these reasons, we cannot publish previously copyrighted maps or satellite images created using proprietary data, such as Google software (Google Maps, Street View, and Earth). For more information, see our copyright guidelines: http://journals.plos.org/plosone/s/licenses-and-copyright.

4.1.    You may seek permission from the original copyright holder of Figures 1, 6, 13 to publish the content specifically under the CC BY 4.0 license. 

4.2.    If you are unable to obtain permission from the original copyright holder to publish these figures under the CC BY 4.0 license or if the copyright holder’s requirements are incompatible with the CC BY 4.0 license, please either i) remove the figure or ii) supply a replacement figure that complies with the CC BY 4.0 license. Please check copyright information on all replacement figures and update the figure caption with source information. If applicable, please specify in the figure caption text when a figure is similar but not identical to the original image and is therefore for illustrative purposes only.

Reviewers' comments:

Reviewer's Responses to Questions

**Comments to the Author**

1. Is the manuscript technically sound, and do the data support the conclusions?

Reviewer #1: Yes

2. Has the statistical analysis been performed appropriately and rigorously? 

Reviewer #1: I Don't Know

3. Have the authors made all data underlying the findings in their manuscript fully available?

Reviewer #1: Yes

4. Is the manuscript presented in an intelligible fashion and written in standard English?

Reviewer #1: Yes

5. Review Comments to the Author

Reviewer #1: INTRODUCTION

In the introduction, it would be useful with some more details on the marine heatwave occurring in 2012. What temperatures occurred, where (bottom, surface, all layers?), and when (spring, summer?).

The first time “Gulf of Maine” is mentioned in the text it should be followed by the abbreviation GOM in brackets, thereafter the abbreviation should be used (exceptions figure and table legends, and abstract). In the introduction, GOM and Gulf of Maine are used interchangeably.

Line 61: remove “)” after GOM

Line 76-77: Re the sentence “age assignments based on length, life history stage and reproductive status”, how can age be determined based on life history stage and reproductive status, as for instance males and berried females may consist of at least two year-classes: I would suggest to just write “age assignments based on length”.

MATERIALS AND METHODS

Line 109-110: Text reads “The inshore survey data included depths from ~40 m to 120 m …” while Table 1 says ~37 m to 120 m. The same interval should be used in text and Table 1.

Line 110: “… spatial overlap analysis, 5-120 m; …” it is not clear what data are used in the spatial overlap analysis, I cannot understand that there exist trawl data from depths of 5 m!

Table 1, last line and first column in Table should be: “ASMFC and NEFSC summer” as it is written earlier in text: “… summer offshore surveys jointly by the Atlantic States Marine Fisheries Commission (ASMFC) and NEFSC.”

The summer offshore survey is a dedicated shrimp survey. Please insert a sentence specifying what the target species of the other surveys are (or perhaps ecosystem surveys?) It says in caption of Fig. 2 that they are “resource” surveys, this should be stated in M&M.

Lines 129-130: In this sentence “NEFSC offshore survey methods were modernized in 2009 with a new survey vessel, a modified trawl net design and shorter tow duration (Table 1; [37])” I would include also “different trawling speed”.

Line 145: should the year interval 1991-2004 include also 2005? As the text states that shrimp was not identified to species for the years 2006-2012.

Line 148: Insert a sentence stating how sampling of shrimp for length and stage determination was carried out on the spring inshore and summer offshore surveys. How many shrimps were sampled and measured per trawl station, and were shrimp sampled at all trawl stations?

Lines 152-153: were the “reference annual cycle” daily means of temperature recorded in the period 1978-1987? I suggest to include a brief description of the “reference annual cycle”

How was sea surface temperature measured? This is not mentioned in the M&M, but sea surface temperature anomalies are shown in Fig. 4.

I assume bottom temperatures from the inshore and offshore summer surveys also are from CTD measurements, this should be explicitly mentioned.

Line 163: Transition to spring and fall: what time period (years) are incorporated in the “average annual temperature”?

Lines 218-219: what is the difference between “bivalves” and “bivalve mollusks”?

Line 231: Bis should be in italics

Lines 242- : Re stomach data sampling, it would be useful with a sentence stating if stomach sampling was carried out for all fish species in the catch.

Line 267: “I” is from Eq. 3, not 2.

Line 273-274: the extra shallow strata in the inshore survey is here said to cover depths between approx. 9 and 37 m, while in line 110, the depth interval is given as 5-37 m. The same interval should be used.

A lot of fish species have been investigated in this work, and it is hard to keep track of which are important where (stomach sampling of new species, stomach sampling of already identified shrimp predators, biomass peaks in 2012, spatial overlap studies). I had to read the M&M and Results section several times to get an overview. Would it be an idea to include a Table in the M&M that gives an overview including number of investigated species per topic, and number of resulting “interesting” species presented in results/Figures, and also a list of the names of these “interesting” species, per topic.

RESULTS

Fig. 2 should have error bars, but I do see the dilemma as these might make the figure less readable as the bars (means) for the years with lowest densities will be even smaller, in order to include error bars for the highest values.

Fig. 3: text in figure is small and very hard to read.

Fig. 4 legend: should include the abbreviations BTA and SSTA. I suggest writing: (A) Bottom and sea surface temperature anomalies (respectively BTA and SSTA) for NEFSC spring and fall surveys

Somewhere there should be included some information on long-finned squid. How large do they grow, i.e. is it likely that they will prey on both male shrimp and the large females? Are they benthic or pelagic or both?

Fig. 7, legend should explain the hatched line = 0.85; that species were selected for further consideration if they had a standardized biomass index above 0.85 in 2012.

Fig. 8, the circles and squares are difficult to tell apart in the figure. I also wonder about the statement: “The only species that was an outlier in terms of percent of stomach content by weight (PW) in 2012 was Atlantic cod in spring”, as I don’t see an outlier in the cod-graph in 2012. The only high frequency (PW) in 2012 seems to be for smooth skate.

Lines 383-385: This sentence talks about 10 fish predators, while Fig 8 shows data from 11 species: “Pandalid shrimp did not occur more frequently in stomach contents (PFO) of the 10 fish predators with sufficient sample sizes during NEFSC spring or fall offshore surveys in 2012 (PFO, Fig 8).”

Line 401: spatial overlap. I wonder why smooth skate was not considered for spatial overlap as this species had rather high percentage of shrimp in its diet (Fig8) as well as higher biomass index in 2012 compared to for instance cod (Fig7), which was included in the spatial overlap analysis.

Fig 11. I wonder how F can be given on an axis from 0 to 20. F is normally a value between 0 and 1.

DISCUSSION

Line 463-465: “estimates were also very high during 1995-2001, exceeding the 2011-2012 values in two of the three models evaluated, with no concomitant collapse of the population.” Here, I think it should be noted that although the population did not collapse, it did however, reach the same low level as in 2012 (Fig.11) (at least the SSB).

Fig 12: this figure should be extended until present. If temperature data are not available, at least the landings data of shrimp and squid should be presented.

GENERAL COMMENTS ON MS, ESPECIALLY DISCUSSION

This is very interesting work, and the issues explored are impressing. The introduction is well written and gives a good overview of the background for the research presented. A lot of species and topics were explored and it is hard to get an overview, thus in the methods section I suggest an overview table. The results are well presented and well discussed. Below are some points for consideration.

The authors talk about the population collapse happening in 2012, but might it be more correct to talk about a collapse happening over the two years 2012-2013, as the really low population size was reached in 2013?

It would be interesting if the authors in the discussion would comment on the fact that the population has remained in a collapsed state also after 2012-2013. And discuss possible causes. Fig 11 shows that the population in 2001 (SSB) was at the same low level as in 2012. However, at that time the population quickly recovered. Why has the population not recovered in recent years? This should be touched upon in the discussion. Does the squid continue to exert a strong predation pressure on the shrimp? Is the bottom temperature still at record high levels? Is spring still occurring earlier than usual? Or is the shrimp population now at such low levels (below a not-specificed Blim-level) that recruitment is impaired?

Information on the recruitment to the stock is missing, other then what can be read from Fig. 3. It would be useful with a plot of the time-series of recruitment (age 1 shrimp). As a short-lived species, the size of the P. borealis stock is highly dependent on the incoming recruitment. Thus, the size of the stock in 2012 was influenced also (in addition to increased predation) by the low recruitment in 2011 leading to low numbers of 2-year old males in 2012.

6. PLOS authors have the option to publish the peer review history of their article (what does this mean?). If published, this will include your full peer review and any attached files.

Reviewer #1: No

---

## [Author Response · Author response to Decision Letter 0]

10 Jun 2021

please see response to reviewers.docx

---

## [Editor Report · Decision Letter 1]

16 Jun 2021

Northern shrimp Pandalus borealis population collapse linked to climate-driven shifts in predator distribution

PONE-D-21-06531R1

Dear Dr. Richards,

We’re pleased to inform you that your manuscript has been judged scientifically suitable for publication and will be formally accepted for publication once it meets all outstanding technical requirements.

Kind regards,

Andrea Belgrano, Ph.D.

Academic Editor

PLOS ONE

---

## [Editor Report · Acceptance letter]

30 Jun 2021

PONE-D-21-06531R1 

Northern shrimp *Pandalus borealis* population collapse linked to climate-driven shifts in predator distribution 

Dear Dr. Richards:

I'm pleased to inform you that your manuscript has been deemed suitable for publication in PLOS ONE. Congratulations! Your manuscript is now with our production department. 

Kind regards, 

on behalf of

Dr. Andrea Belgrano 

Academic Editor

PLOS ONE